# The Methodology for Assessing the Applicability of CSR into Supplier Management Systems

**Ferdinand Kóča** [1], **Hana Pačaiová** [1,*], **Renata Turisová** [1,*], **Andrea Sütőová** [2] and **Peter Darvaši** [1]

[1] Faculty of Mechanical Engineering, Technical University of Košice, Letná 1/9, 04200 Košice, Slovakia; ferdinad.koca@student.tuke.sk (F.K.); peter.darvasi@student.tuke.sk (P.D.)

[2] Faculty of Materials, Metallurgy and Recycling, Technical University of Košice, Letná 9, 04200 Košice, Slovakia; andrea.sutoovoa@tuke.sk

[*] Correspondence: hana.pacaiova@tuke.sk (H.P.); renata.turisova@tuke.sk (R.T.); Tel.: +421-903-719-474 (H.P.)

**Abstract:** The implementation of management systems has become a strategic advantage in achieving business goals, especially in industrial organizations, but the implementation of social responsibility requirements is an especially ethical issue. Due to the existence of various standards (often industry-specific) as well as individual codes of conduct developed by large multinational organizations, supplier organizations must face a variety of requirements. The question, then, is to what extent their established management systems (MSs) meet these requirements. The objectives of the study were to: (1) analyze the different CSR requirements of internationally recognized cross-industry and industry-specific standards and codes in different industries; (2) select the most appropriate framework and develop a methodology for assessing the degree of applicability of CSR in the selected management systems; (3) apply the proposed methodology (the so-called Social Requirements Applicability in Management Systems—SRIMS) in the selected areas: automotive industry, research organization, and metallurgical industry; and (4) analyze the results of SRIMS by the application of the ANOVA and Bonferroni method and define clusters within the selected factors—"Organization", "Standard", and "Chapter" and determine differences between pairs within each factor. The application of the Bonferroni method confirmed the hypotheses that the developed SRIMS model is an appropriate tool for assessing the overall level of applicability of CSR requirements in established MSs.

**Keywords:** corporate social responsibility (CRS); sustainability; management systems (MSs); ANOVA; integration of management systems (IMS); suppliers; applicability

## 1. Introduction

The globalization of economics has created opportunities that lead to an ever-increasing number of suppliers of raw materials and products. Growing stakeholder interests, more complex business processes, and relationships have created pressure to develop and integrate socially responsible practices across global and local industries [1,2]. Organizations are increasingly required to balance the social, economic, and environmental domains of their business while increasing the organization's value for their shareholders. Corporate social responsibility is not just about the organizations themselves, but is also about the entire supply chain [3–5].

Today, large multinational organizations must be responsible not only for the environmental impacts on their global business partners (e.g., suppliers, logistics providers, and intermediaries) but also for their approaches to employee management and care [6]. Any abusive or illegal treatment of employees within the organization itself or in its supply chain can damage its reputation. According to Hofmann [7], examples of organizations where undesirable practices have been found in their supply chain are: Zara, Apple, and Nestle's KitKat. CSR management in the supply chain can enhance the image and opportunities of an organization. Sustainability is an essential component of a competitive advantage

for organizations and for preserving the possibilities of future generations to meet their needs. The application of CSR principles significantly supports sustainability and is a tool contributing to achieving the goals of sustainable development [8–11].

Social justice, a healthy working and living environment, maintaining relationships with customers and other interested parties, and a responsible business approach emphasize a holistic style of management to increase performance.

Developments in the field of management systems required a change from isolation in the given area (e.g., quality, environment, and safety) to their integration on the basis of risk. This integration creates a core support and linkage with the CSR requirements.

CSR management in the supply chains can increase risk resilience and improve the organization's image and opportunities [7].

Many corporate companies develop and implement their own codes of conduct, which are usually supported by sophisticated management systems (MSs), and in many cases, they also develop codes for their suppliers to demonstrate their CSR commitments and to meet the demands placed on them by their stakeholders. Codes of conduct are often based on local laws, international agreements and standards and they are complemented by the organizations' own CSR strategies and priorities [12]. The end customer sets the principles that suppliers must adhere to and must demonstrate that their activities are consistent with them if they want to successfully maintain a business relationship [13,14]. In addition, the organization can ask its suppliers for certificates to verify compliance with, for example, environmental, safety, and social requirements. Such policies provide valuable criteria for decision-making in the selection process and supplier evaluation, as well as for self-improvement of supplier performance in the supply chain [15–17]. Compliance with requirements can be verified using various self-assessment and auditing approaches [18].

To standardize CSR requirements, several universal and sector-specific international standards have been developed (e.g., in the electrotechnical, automotive, raw material extraction, agricultural, construction, apparel, and other industries) with schemes for implementation, monitoring, and certification. The advantage of using universal CSR standards in organizations is that it reduces the burden on suppliers to apply and comply with them. They can also prevent inefficiencies in management and in the prevention of non-conformities [19]. However, many end-user organizations adapt these standards to their values and business objectives or create their own codes [20,21]. Hence, suppliers can often be confronted with different CSR requirements from different customers.

There are many sources [3,6,7,22–24] addressing the issue of CSR in the supply chain, but only a few studies have dealt with the problem of the diversity of demands placed on suppliers resulting from different standards and codes applied by end-user organizations [25,26]. Also, there are no studies exploring the level of compliance with CSR requirements in management systems (MSs). Only a few studies have focused on the examination of the effect of selected MSs on the field of CSR activities [27,28].

The objective of this research was to identify the issues arising in supplier organizations in relation to CSR compliance demonstration with different requirements during second or third party audits or self-assessment processes and to propose a methodological framework (the so-called Social Requirements Applicability in Management Systems—SRIMS) for assessing the level of fulfilment of CSR requirements in established MSs. The proposed methodology was verified in selected organizations operating in the fields of research; automotive and metallurgy having differently established management systems. The next goal was to analyze the results from SRIMS evaluation in more detail by the application of the ANOVA and Bonferroni method to determine the suitability of the proposed SRIMS methodology and to identify similarities within the analyzed factors—the organization (that was involved in the study), the standard (implemented ISO MSs) and the chapters (chapters of ISO standards containing requirements).

## 2. Theoretical Framework

### 2.1. CSR Requirements in the Supply Chain

EU trade agreements now also include rules on social responsibility, in the areas of compliance with environmental, labor, and legal standards [29–31].

In fact, the very concept of corporate social responsibility (CSR) has evolved from a philanthropic approach to today's strategic business imperative for organizations to achieve a competitive advantage [20,32,33]. Despite many efforts to provide a clear and unbiased definition of CSR, there is still ambiguity in both the business and academic worlds as to how CSR should be clearly defined. Behringer [34] came to the conclusion that CSR is a business model that promotes business contributions to sustainable development, i.e., it strikes a balance between economic, environmental needs, and ethical concerns. Schwart [35] introduced the so-called "Three Domain Model for CSR", which consists of three basic domains depicted in the shape of circles: economic, legal, and ethical. The model suggests that none of the three domains of CSR is more important or more significant than the others and their application should be equally balanced. The ideal overlap for CSR lies in the middle of the three circles of the model, at the intersection of all three domains where economic, legal, and ethical responsibilities are simultaneously fulfilled in the organization. It can now be said that the legal and ethical domains overlap, and the environmental domain has been brought to the fore, with the legal domain being indirectly applied across all three domains [35–37].

The most common forms of ensuring social responsibility in supply chains are standards and codes of conduct, the application and monitoring of which are then used to assess compliance and evaluate an organization's performance in the area of CSR. According to Yawar [38], the social domain in CSR is not immutable and depends on many factors, such as culture, trust among stakeholders, organizational strategies, and others, which can be effectively managed through continuous dialogue with stakeholders and mutual understanding of the most important social requirements in the supply chain.

Just as the requirements for suppliers regarding CSR have evolved in recent years, so have the requirements of customers for the implementation and certification of the management systems they require from suppliers. There are now a number of management systems (MSs) with different focuses, standardized according to international standards, e.g., Quality Management, Environmental Management, Occupational Health and Safety Management, Energy Management, Information Security Management, Food Safety Management, Anti-corruption Behavior Management Systems and many other standards and guides for different sectors [39–42].

ISO standards can interact with each other, i.e., they can be combined and integrated. Organizations that already use a standard for a selected management area can implement other areas in an easier way. This is due to their harmonized structure known as the "Harmonized Approach for Management System Standards". The principle of integration is set out by the SL Annex, the so-called "High-Level Structure" (HLS) [28,42–47].

Reflecting on CSR requirements, we can conclude that many of the above-mentioned standards already help to partially meet some of the CSR requirements, but many times this is still not sufficient. As stated by Zhang [48], good CSR performance can enhance an organization's credibility, strengthen its relationships with stakeholders and create a good reputation for the organization. Customer organizations use two main ways to evaluate and monitor supplier performance; those are auditing or self-assessment [49–51]. We can say that both methods are often directly or indirectly coercive strategies for suppliers to meet environmental, ethical, and economic requirements.

As Bajwa [52] states in planning and conducting supplier audits, thanks to the so-called blockchain, easily accessible and transparent supplier data can be used to make more correct decisions about which suppliers to audit, and how and where to focus the efforts and resources needed to conduct audits. Stakeholder pressure, cooperation, and supplier development (e.g., training and education), as well as the increase in ICT development can

provide further opportunities to improve supplier organizations' performance in particular CSR areas.

In addition to auditing suppliers to verify the CSR practices in place, supplier self-assessment through a questionnaire is a frequently used tool, especially for global purchasing companies.

Fraser [53] analyzed supplier sustainability self-assessment questionnaires and concluded that they are one of the most-common tools used in supply chain sustainability management in almost all industries. Many industry initiatives, such as Drive Sustainability, the automotive industry peer group and its self-assessment questionnaire "Sustainability Assessment Questionnaire (SAQ)", the electronics industry citizens coalition (EICC), and the ethical toy program (IETP) in the toy industry, continue to develop common and standardized questionnaires and sustainability-related standards, in the supply chain [51–56].

Sustainability in supply chain management (SSCM) according to Yawar [38] is, "The management of material, information, and capital flows, as well as collaboration between companies within the supply chain, taking into account objectives from all three dimensions of sustainable development i.e., economic, environmental and societal, which are derived from customer and stakeholder requirements". At the top, the scheme for assessing supplier sustainability in the context of organizational performance [16,45] describes six core areas (environment, social values, ethics, economic stability, operational performance, internal impacts, and external impacts), which, when broken down, make up a total of eighteen items for assessing sustainability in CSR. At the bottom, the scheme is supplemented by auditing and evaluation as a separate process for assessing suppliers by auditing and/or a self-assessment questionnaire.

Many organizations are focusing on blockchain implementation to facilitate transparency in product lifecycle, circular economy, and supply chains, and to better control their environmental footprint [56]. According to Bajwa [52], the use of blockchain minimizes the amount of redundant data because all information is entered only once and is viewed by all who need it.

The blockchain system [52,54–56] is a technology that enables data traceability, a way of identifying business requirements and data from the perspective of the relevant organizations at the end of the chain, for transactional data when goods change ownership in the supply chain. Blockchain technology has two important aspects, and these are a database to record transactions physically stored in multiple copies, in different locations, and a system of "trust" between different users, enabling and requiring them to give consensual and digital consent to any changes in the database [56].

*2.2. CSR Standards Framework*

A study focusing on existing universal and selected industry CSR standards and codes of ethics was described in the paper by Sütőová [25]. It was divided into three parts: a review of universal standards, the standards, and the requirements in the electrotechnical and automotive industries.

The social responsibility management system is described by a single certification standard, IQNet SR10 [57], which is based on the principles and recommendations of ISO 26000 [58]. Although this standard provides guidance on CSR, it is not intended for certification [59,60].

There are also other initiatives creating principles and standards for the reporting of sustainability impacts of the organization's activities, e.g., GRI (Global Reporting Initiative). It is advantageous if CSR reports are provided by an independent organization to objectively assess compliance in supplier organizations and reduce information risk in communication [61].

In addition to the above-mentioned cross-industry standards and codes (applicable to organizations of all types and sizes), industry-specific codes are used to regulate negotiations between industry participants. Codes developed and used by individual organizations may also regulate relationships between customers and suppliers.

Table 1 provides an overview of the codes and assessment frameworks used in the electronics (EL), automotive (AU), and steel (ST) industry and the individual codes of the organizations cooperating within these industries.

**Table 1.** Overview of CSR codes and assessment frameworks used in selected industries.

| CSR Standards and Codes | Basic Subject and Requirements | Approaches to Assessment | EL | AU | ST |
|---|---|---|---|---|---|
| RBA (Responsible Business Alliance) 2020 [62] | RBA Code of Conduct: 1. Staff; 2. Health and Safety; 3. Environment; 4. Ethics; and 5. Management systems | RBA VAP, auditable by a third party. | √ | | |
| Electrolux Supplier standards in the workplace 2020 [63] | Child labor; Workforce; Safety measures; Health and safety; Non-discrimination; Harassment and abuse; Disciplinary and grievance procedures; Working time; Compensation; Freedom of assembly; Environmental compliance; and Corruption and business ethics. | Electrolux Workplace Policy and Supplier Workplace Standard (second- and third-party audits). | √ | | |
| BSH Supplier Code of Conduct 2021 [64] | Laws and regulations; Corruption and bribery; Human rights; Labor; Child labor; Harassment; Compensation; Hours of work; Non-discrimination; Health and safety; Freedom of assembly and collective bargaining; Environment; and Supply chain. | CSR audit carried out by a third party towards BSH Supplier Code of Conduct. | √ | | |
| IATF 16949 [65] | CSR policy that, as a minimum, should include: Anti-Bribery Policy, Employee Code of Conduct, and Ethics Escalation Policy. | | | √ | |
| SAQ ver. 5.0 2021 [66] | Business Management, Working Conditions and Human Rights, Health and Safety, Business Ethics, Environment, Supplier Management, and Responsible Sourcing of Raw Materials. | | | √ | |
| BMW Group Supplier Sustainability Policy 2021 [67] | 1. Environmental responsibility; 2. Social responsibility; 3. Public governance; and 4. Supply chain responsibility | SAQ 5.0/RBA VAP (third party audit) | | √ | |
| FORD Human Rights Code, basic working conditions social responsibility 2022 [68] | 1. Human rights and working conditions; 2. Community involvement and indigenous peoples; 3. Bribery and corruption; 4. Environment and sustainability; and 5. Accountability and implementation | SAQ 5.0/RBA VAP (third party audit) | | √ | |
| PSA Group Responsible Purchasing Rules [69] 2020 | 1. Social principles; 2. Environmental protection; 3. Ethical principles; and 4. Sustainable procurement | EcoVadis Platform/PSA Group Own methodology (third party audits) | | √ | |

**Table 1.** *Cont.*

| CSR Standards and Codes | Basic Subject and Requirements | Approaches to Assessment | EL | AU | ST |
|---|---|---|---|---|---|
| Volkswagen Group Code of Conduct for Business Partners 2020 [70] | 1. Environmental protection; 2. Human and labor rights of employees; 3. Transparent business relations; 4. Fair market conduct; 5. Due diligence to promote a responsible mineral supply chain; and 6. Integration of sustainability requirements in the organization and processes. | SAQ 5.0 / RBA VAP (third party audit) | | √ | |
| FCA Group Sustainability guidelines for suppliers 2020 [71] | 1. Human rights and working conditions; 2. Environment; 3. Business ethics and corruption; and 4. Monitoring and corrective action. | RBA (by third party towards Supplier Code of Conduct) | | √ | |
| ResponsibleSteel 2021 [72] | 1. Company Management; 2. Social, Environment and Governance Management System; 3. Responsible Sourcing of Input Materials; 4. Decommissioning and Closure; 5. Occupational Health and Safety; 6. Labor rights; 7. Human Rights; 8. Stakeholder Engagement and Communication; 9. Local communities; 10. Climate Change and Greenhouse Gas; 11. Noise, Emissions, Effluents and Waste; 12. Water Stewardship; and 13. Biodiversity. | Third party audit according to ResponsibleSteel standard. | | | √ |
| ThyssenKrupp Supplier Code of Conduct 2020 [73] | Human and labor rights; Employee health and safety; Environmental protection; Business conduct; Supplier relations; and Compliance with the ThyssenKrupp Code of Conduct. | ThyssenKrupp Supplier Code of Conduct (second- or third-party audit) | | | √ |

Many organizations operating in the electronics industry (including leading companies, such as LG, Samsung, BSH-Siemens, etc.) have adopted a common code of conduct developed by the Responsible Business Alliance (RBA). It is the world's largest industry coalition focused on corporate social responsibility in global supply chains. The RBA criteria are also a condition of cooperation for suppliers of these organizations [62]. RBA members are predominantly companies operating in the electrical industry, but this does not mean that it is not applicable to other industries. Third party audits are conducted by RBA member affiliates and their supplier affiliates in accordance with the RBA Code of Conduct under the name Validated Assessment Program (VAP) [74]. Despite the existence of the RBA, some organizations in the electrical sector have their own codes of conduct or have implemented other standards reflecting their values and priorities, and their suppliers must comply with these codes and standards.

IATF 16949, an industry certification standard used in the automotive supply chain, includes requirements for social responsibility in Chapter 5.1.1.1. These requirements appeared in the latest revision of the standard published in 2016. The requirements defined

by the standard are the establishment of a social responsibility policy regarding bribery, rules of conduct for employees, and the escalation of ethics.

In an effort to unify CSR requirements for suppliers in the automotive industry, the Drive Sustainability partner group consisting of 18 leading automotive organizations (BMW Group, Daimler Truck, Ford, Geely, Honda, Jaguar Land Rover, Mercedes-Benz, Scania CV AB, Toyota Motor Europe, Volkswagen Group, Volvo Cars, Volvo Group and Ferrari, GWM, Polestar, Stellantis, UD Trucks, and Volta Trucks) has created a common Sustainability Assessment Questionnaire (SAQ), a questionnaire that is regularly revised [45]. In 2022, the fifth version of the questionnaire (SAQ 5.0) was published and is used by many automotive companies to assess supply chain sustainability, including sourcing, manufacturing, assembly, and logistics. The SAQ is aligned with the Global Automotive Sustainability Guiding Principles (GASGP) set by Drive Sustainability and the AIAG partner group. The GASGP include expectations for suppliers on key responsibility issues, including human rights, environment, working conditions, business ethics, health and safety, and responsible supply chain management.

Organizations from each stage of the steel supply chain have created an independent certification standard and program known as ResponsibleSteel, which was first published in late 2019. A revised version of the standard—ResponsibleSteel International Standard V2.0—was published in 2022. The 13 principles of the Standard cover environmental, social, and governance issues, which were identified and agreed upon by members and stakeholders. AcelorMittal is the driving force behind the creation of the ResponsibleSteel program, along with other steel producers, such as Voestapine, Blue Scope, Aperam (United States Steel will be added in 2021), and some OEMs, such as Daimler and BMW, and civil society organizations [25].

## 3. Materials and Methods

Based on the review of standards and approaches to CSR, the IQNet SR10 standard was chosen as the basis for the development of a methodology enabling the assessment of the level of applicability of CSR in the selected MSs. This standard is compatible in its structure with other management system standards (according to ISO Annex SL) and is also intended for auditing and certification. The research team compared the CSR requirements of IQNet SR10 to the following management systems: ISO 9001 (QMS) [75], IATF 16949 (IATF) [65], ISO 14001 (EMS) [76], ISO 45001 (OHSMS) [77], and ISO 50001 (EnMS) [78], which can be considered as an Integrated Management System (IMS) if implemented simultaneously in an organization (see Figure 1), to create a basic framework for the model.

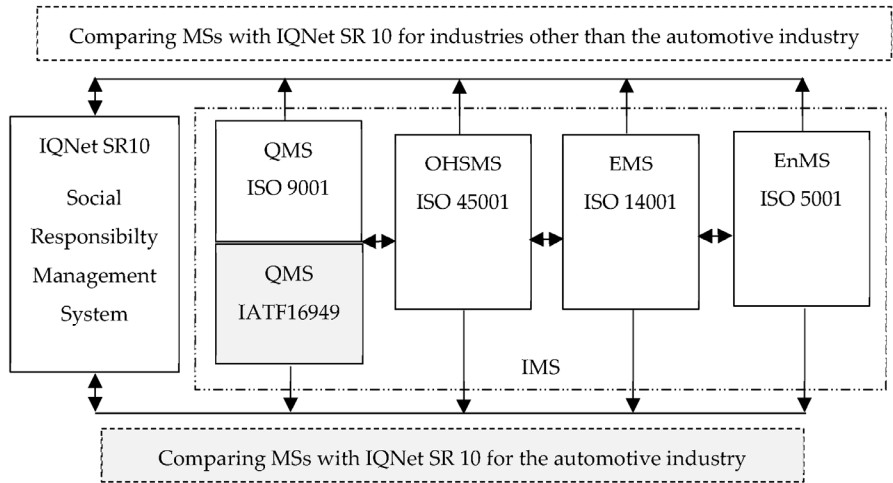

**Figure 1.** Framework for a methodology to assess the applicability of CSR in MSs (own processing).

The requirements of the chapters of the selected management systems were compared with the structure of the requirements described in IQNet SR10. The development of the methodological framework for assessing the applicability of CSR requirements to MSs was based mainly on the structural and subject matter consistency of the requirements.

The SRIMS (Social Requirements Applicability in Management Systems) was created for organizations that need to implement social responsibility in their organization and to evaluate their level of implementation in order to further improve their processes effectively.

In the development of the methodology, the sectoral different approaches were also taken into account, i.e., if the organization was a supplier to the automotive industry, the conformity check was applied by assessing the integration in quality management through the comparison of IATF 16949 requirements, otherwise only the ISO 9001 approach was applied (see Figure 1).

Depending on the structure of the sub-chapter requirements, if the weight $W_X$ of the chapter was conditioned on compliance based on the conditional compliance of the performance of its sub-chapters $W_{XY}$, where $Y = 1, 2, \ldots, m$, then the value of the weight of every $X$-chapter ($X = 4, 5 \ldots, 10$) reflected the sum of the weights assigned to the individual $Y$-sub-chapters:

$$W_X = \sum_{Y=1}^{m} W_{X.Y}; \; W_{X.Y} \geq 0, \tag{1}$$

The calibration of the methodology was chosen according to the following criteria: for each sub-chapter ($W_{X.Y}$), 20 points could be achieved. In the case of a large difference in the number of sub-chapters $Y$ in a given Chapter $X$ (e.g., Chapter 10, $X = 10$), its importance in the overall structure of the MS was taken into account (similarly to the EFQM model) [79]. Therefore, a double value could be achieved for sub-chapter $W_{X.Y}$, a maximum of $2 \times 20$ points (see Table 2). Clearly, not every MS achieves the same total score (see Appendix A, Table A1).

**Table 2.** Assignment of weights (calibration of the SRIMS methodology).

| Standard Chapters/($W_X$) | | Management System (MS) | | | | | |
|---|---|---|---|---|---|---|---|
| | | IQNet SR10 | QMS | IATF | EMS | OHSMS | EnMS |
| 4 | Context of the organization | 80 | 80 | 80 | 80 | 80 | 80 |
| 5 | Leadership | 80 | 60 | 80 | 60 | 80 | 60 |
| 6 | Planning | 80 | 60 | 60 | 60 | 60 | 120 |
| 7 | Support | 100 | 100 | 100 | 100 | 100 | 100 |
| 8 | Operation | 190 | 150 | 180 | 40 | 60 | 60 |
| 9 | Performance evaluation | 100 | 80 | 80 | 80 | 80 | 80 |
| 10 | Improvement | 80 | 80 | 80 | 80 | 80 | 80 |
| | In total $W_X$ | 710 | 610 | 660 | 500 | 540 | 580 |

The application of the methodology, known as SRIMS, has been verified in plants that have long-established management systems. These were:

- A research organization (TU), focused on the development of electronic systems, which had an ISO 9001 QMS in place, but other MSs were not applied, even though it must comply with other requirements of its customers in its activities.
- Three organizations that are suppliers to the automotive industry. Two of them, AU1 and AU2, did not have an Energy Management System (EnMS) in place, but had a system for CSR according to IQNet SR10. Only AU3 had EnMS. It also had CSR requirements in its policy but IQNet SR10 was not implemented.
- The last respondent for model verification was a metallurgical company which also had an EnMS in place, but its CSR policy was not compliant with IQNet SR10.

An overall summary of the respondents subjected to the SRIMS integration survey, with respect to the management systems implemented, is provided in Table 3.

**Table 3.** Respondents of SRIMS.

| Respondent | | Management System (MS) | | | | | | |
|---|---|---|---|---|---|---|---|---|
| | | IQNet SR10 | QMS | IATF | EMS | OHSMS | EnMS | Area of Activity |
| 1 | TU | | √ | | | | | EL |
| 2 | AU1 | √ | √ | √ | √ | √ | √ | AU |
| 3 | AU2 | √ | √ | √ | √ | √ | | AU |
| 4 | AU3 | | √ | √ | √ | √ | | AU |
| 5 | OC | | √ | √ | √ | √ | √ | ST |

After the verification of SRIMS methodological framework in the above-mentioned organization, the results from the evaluation were further analyzed by using the ANOVA and Bonferroni method to determine the suitability of the proposed SRIMS methodology and identify clusters within the analyzed factors—organization, standard and chapter and finding similarities by pairwise analysis.

## 4. Results

### 4.1. Comparison of CSR Requirements with Requirements of Selected Management Systems

A comparison of CSR requirements (based on the IQNet SR10 standard) with the management systems creating the integrated management system [64,65] was performed and the following facts were found within individual chapters:

Chapter 4: "Context of the organization" presents a 100% compliance rate of applying IMS requirements against the IQNet SR10 standard and no chapters of the IMS standards were moved to another IQNet SR10 chapter.

Chapter 5: "Leadership". It can be noted that there is only one difference in the IATF 16949 standard intended for the automotive sector, which is quite interesting and groundbreaking regarding CSR, compared to other standards in the IMS framework. This is Chapter 5.1.1.1 "Social Responsibility", which calls for the introduction of a requirement for social responsibility, the introduction of a policy aimed at an anti-bribery employee code of conduct, and an ethics escalation policy. The ethics escalation is the so-called imaginary policy of drawing attention to negative phenomena, i.e., a policy for "whistle-blowing". These requirements are new in the IATF standard (after the 2016 edition) and unique in quality management standards, although these requirements do not include all the policies relevant to CSR. It could be noted that the IATF 16949 standard, with the above-mentioned chapter, greatly helps to focus on CSR in addition to the quality requirements.

Chapter 6: "Planning". After the comparative analysis of Chapter 6 of IQNet SR10 standard versus the ISO standards, it was possible to conclude that there are no major changes in the requirements. For the individual MSs, the requirements for objectives and planning to achieve them are shifted from Chapter 6.2 to Chapter 6.3, which has no impact on the integration of CSR with the other IMS standards. In addition, the IQNet SR10 standard includes Chapter 6.2 Identification and Evaluation of Issues, which contains the positive or negative impacts of stakeholders, taking into account economic, environmental, social impacts, and good organizational governance that affect the organization's sustainability and social responsibility. Unlike other ISO standards, only the publishers of the ISO 50001 standard have included in this chapter the establishment of requirements for defining energy indicators and a baseline for assessing energy use, which makes this standard specific. After considering Chapter 6, one of the three main chapters of all the IMS standards considered was moved in terms of subject matter to the IQNet SR10 standard; this was Chapter 6.2. Sub-chapter 6.1.3 related to The Environmental Management System (EMS) and Occupational Health and Safety Management System (OHSMS) was also moved, which may help to meet legal requirements.

Chapter 7: "Support". After reviewing Chapter 7, it was possible to conclude that there was a 100% compliance rate in applying the IMS requirements against the IQNet

SR10 standard, and no chapters of the IMS standards were moved to another IQNet SR10 chapter. Some of the IMS standards even specify their requirements in more detail.

Chapter 8: "Operation". The main difference between IQNet SR10 and the other ISO standards mentioned above was evident in this chapter. Basically, Chapter 8.1 Planning and Management of Operations is the same for all standards. However, it can be stated that the most important and fundamental requirements for CSR are defined in Chapter 8 of IQNet SR10. These are included in Sub-chapters 8.2 to 8.9. In a more detailed analysis of each chapter, we found that Sub-chapters 8.2 Owners and Stakeholders, 8.6 Government, Public Authorities and Regulators, 8.7 Community, Society and Social Organizations, and 8.9 Competition are completely new requirements that are not supported by other ISO standards, thus these requirements need to be implemented in the organization.

Other sub-chapters of Chapter 8, such as 8.3 Employees, 8.4 Customers, Users and Consumers, 8.5 Product Suppliers, Service Providers and Partners, and 8.8 Environment are partially supported by other ISO standards, but the CSR requirements in the IQNet SR10 standard are more detailed and are linked to the requirements of meeting the global standards of the International Labor Organization, United Nations (ILO).

When analyzing Sub-chapter 8.3.4 Health and Safety, it was possible to declare that if an organization has an Occupational Health and Safety (OHS) management system in place according to ISO 45001, this fundamentally addresses the area of OHS which is part of the CSR requirements. Similarly for Sub-chapter 8.8 Environment, where it can also be stated that if the organization has an environmental management system in place according to ISO 14001, this fact substantially addresses the EMS area which is part of the CSR requirements.

When analyzing the requirements of Sub-chapters 8.3.1, 2, 3, 4 and 8.3.6 of IQNet SR10, it can be pointed out that the implementation of the requirements of SA 8000, which focuses on human and labor rights, can fundamentally help to ensure the organization's compliance with these requirements.

The requirements of IQNet SR10 in Sub-chapter 8.3.5 Accessible working environment help to fulfil the requirements of the Quality-Management-System (QMS)-focused standard by Sub-chapter 7.1.4 Environment for the operation of processes.

In particular, the analysis took into account the subject matter comparison of the requirements and chapters of the standards, leading to the conclusion that in further developing the proposed methodology, it would be worthwhile to consider moving some of the requirements of the IMS standards to another chapter of the IQNet SR10 standard, such as 7.1.2 Workers from the QMS standard to Chapter 8.3 in the case of IQNet SR10. Similarly, Chapter 8.3 Design and development of products and services from the QMS standard could be moved to Chapter 8.4.6 in the case of IQNet SR10. In principle, however, this does not affect the final assessment of the level of compliance.

Chapter 9: "Performance evaluation". When analyzing Chapter 9 of the IQNet SR10 standard, an additional requirement for performance evaluation of the organization was identified, and that is to monitor information related to stakeholder perceptions in Chapter 9.2 on a regular basis in a documented manner as part of the IQNet SR10. Chapter 9.3 "Internal audits" was moved from Chapter 9.2 to 9.3 compared to other ISO standards for IMS. This is similar to Chapter 9.4 "Managerial Review" which is defined in the ISO standards in Chapter 9.3 and in IQNet SR10 in Chapter 9.4. This is due to the decision of the publisher of the standard to insert the Chapter Stakeholders' expectations under 9.2, which has consequently shifted the other chapter assignments, but this does not change the requirements and IQ Net SR10 introduces this essential requirement in Chapter 9 against other ISO standards for IMS.

Chapter 10: "Improvement". When analyzing Chapter 10 of IQNet SR10, it was possible to conclude, as with Chapters 4 and 7, that there are no extra requirements over the other ISO standards for IMS. All the standards mentioned above define two main areas of requirements and these include: Nonconformity and Corrective Action and Continuous Improvement. The difference is the numbering of the chapters in IMS standards, but this has no impact on the integration of CSR with the MSs standards. The IATF 16949

standard defines more explicit requirements for the control of nonconforming outputs, the elimination of nonconformities and preventing the recurrence of problems, and for focusing on customer complaints, which methodologically can only be beneficial for an established IMS in an organization [80–83].

### 4.2. Results of the Evaluation Using SRIMS Methodology in the Selected Organizations with Different Management Systems

The analysis was carried out in cooperation with the managers responsible for MSs in each organization, e.g., the quality manager, the occupational health and safety (OHS) manager, the manager for the integrated management system, and the SRIMS project research team. Each respondent assigned weights to each requirement of the SRIMS methodology (processed in Excel) based on his/her experience. Firstly, they assessed the effectiveness of the required compliance of the implementation of the MS in the given area (referred to as the *MZ* variable), then evaluated whether the MS requirements are consistent with the IQNetSR10 requirements and the extent to which they are applicable in their organization (referred to as the *MU* variable—Rate of applicability). The two variables were summed, and the *Total* variable was determined, characterizing a comprehensive approach corresponding to the CSR applicability rate within MSs in place.

The results of the examination according to SRIMS in each organization were processed in Tables A2–A6 (Appendix B) and graphs (see Figure 2).

The first research organization (with only a QMS implemented) declared a total applicability compliance in SRIMS of only 38% (see Figure 2a), with the lowest scores in *MU* compliance and *MU* applicability in Chapters 6, 8, and 10 (see Figure 2a).

The AU1 organization declared a total applicability fulfilment in SRIMS for MSs: IATF at 88%, the lowest score in Chapters 8 and 10; in EMS 67%, the lowest score in Chapter 10; and in OHSMS 71%, the lowest score of applicability in Chapters 8 and 10 (see Figure 2b).

The AU2 organization declared an overall fulfilment of applicability according to SRIMS for MSs: IATF at 86%, where the lowest score was in Chapter 10; in EMS 64%, the lowest score was in Chapter 10; in OHSMS 69% where the lowest scores were in applicability and compliance in Chapters 8 and 10 (see Figure 2c).

The AU3 organization declared an overall SRIMS applicability performance for MSs: IATF at 86%, with the lowest score in Chapter 10; in EMS 65%, with the lowest score in Chapter 10; in OHSMS 69%, with the lowest score for applicability and compliance in Chapters 8 and 10; and in EnMS 70%, in Chapters 6 and 10 (see Figure 2d).

The metallurgical organization declared an overall applicability performance in the SRIMS model for MSs: IATF of 77%, the lowest score in Chapters 4, 5, and 8; in EMS 55%, the lowest score in Chapters 4, 5, and 10; in OHSMS 61%, the lowest score in applicability and compliance in Chapters 4, 7, 8, and 10; and in EnMS, 62%, the lowest score in Chapters 5, 4, and 10 (see Figure 2).

As the SRIMS methodology needed to evaluate more than one variable simultaneously, a multifactorial technique was applied in the next step to assess the relevance of the results obtained.

### 4.3. Results of ANOVA and Bonferroni Analysis

For validation, a multi-factor ANOVA analysis (Analysis of Variance) was used, which is widely applied as a powerful parametric statistical technique [84–88].

The individual factors represented categorical explanatory variables: the first factor was called "Organization", which were the organizations involved in the survey: TU, AUT1, AUT2, AUT3, and OC, according to the enterprises in which the survey was conducted. We were interested in the differences between the enterprises in the requirements for the implementation of MSs.

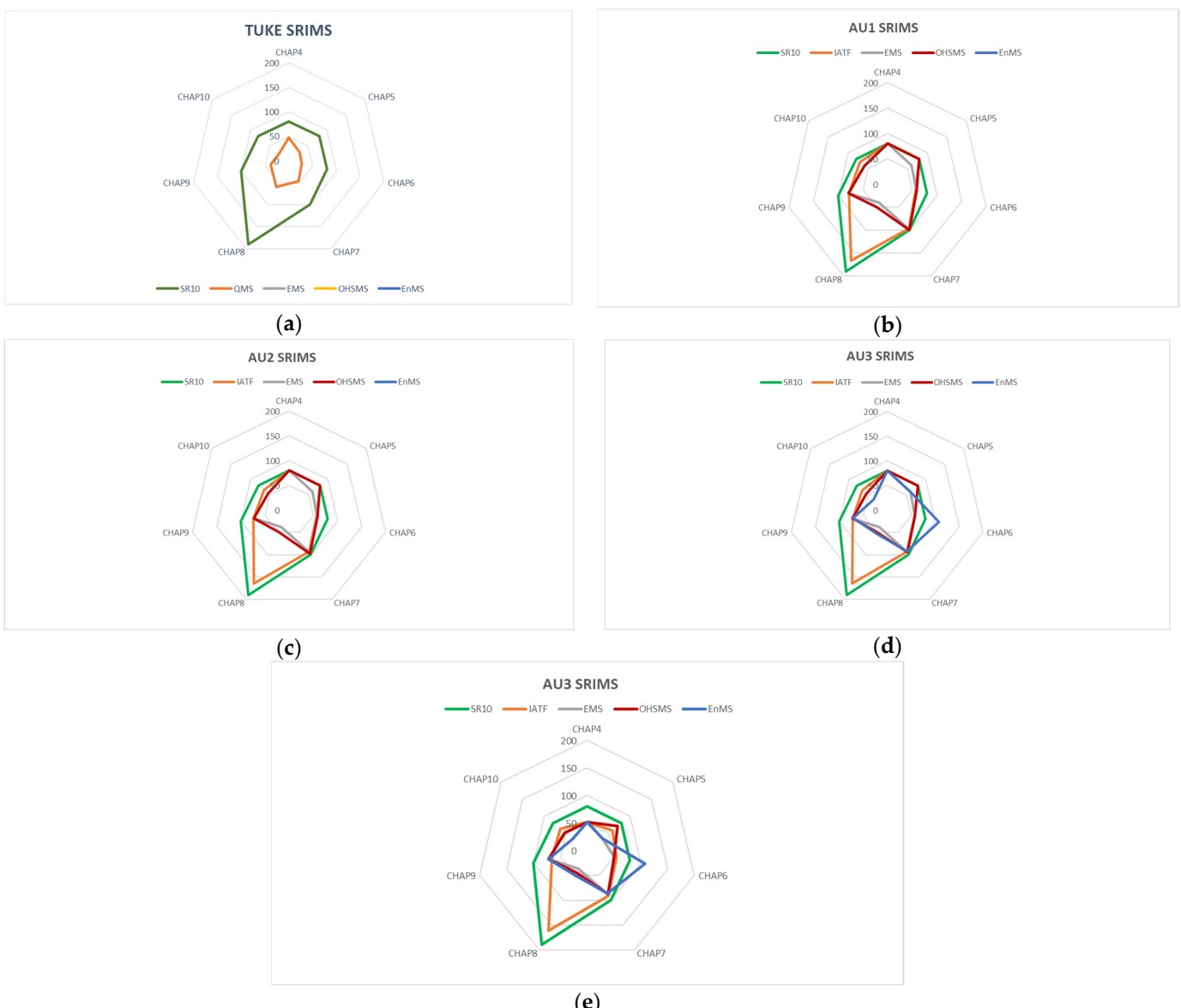

**Figure 2.** A spider diagram assessing the overall level of applicability of CSR in established MSs among SRIMS respondents: TU (**a**); AU1 (**b**); AU2 (**c**); AU3 (**d**); and OC (**e**) (own processing).

The "Standard" was chosen as the next factor of the experiment. This factor also has five levels, according to the evaluated compliance requirements in MSs, i.e., for quality (QMS), automotive (IATF), safety (OHSMS), environment (EMS), and energy management (EnMS). For this factor, we were interested in the degree of variation in the level of implementation of the requirements of each MSs across the queries.

The third factor—"Chapter"—related to the mentioned standards and the assessment of the level of integration of CSR requirements. Individual Chapters (specifically Chapters 4 to 10) oriented each other equally across all the standards considered. This factor had seven levels, namely from Chapter 4 to Chapter 10 (see Table 3).

A general linear model was used. Based on the nature of the data, this was an unbalanced ANOVA model. As responses (or independent variables or explanatory variables) we used: a variable referred to as *MZ* (compliance rate) with the requirements of the established MSs, a variable *MU* (rate of applicability) and a parameter referred to as *Total* (rate of overall applicability of the CSR requirements in the MSs), which numerically is the sum of *MZ* and *MU*. The specific values obtained in SRIMS were used. The maximum values for each weight served as the basis for the scores (see Table 2). 3 Independent

expert assessments of the SRIMS were conducted in each organization by the involved top managers of each organization.

The ANOVA method was used to analyze the results, followed by the Bonferroni method as a post-ANOVA analysis. Minitab software was used to evaluate the ANOVA method. Table 4 presents the designation of each factor level that was used in further analysis.

**Table 4.** Designation of each factor level.

| Factor | Level | Values |
|---|---|---|
| Organization | 5 | AUT1; AUT2; AUT3; OC; TU |
| Standard | 5 | EMS; EnMS; IATF; OHSAS; QMS |
| Chapter | 7 | 4; 5; 6; 7; 8; 9; 10 |

In the following, the conditions under which the ANOVA method can be used were verified. Tables 5 and 6 present the verification of homoscedasticity for both *MZ* and *MU* responses (variables).

**Table 5.** Test for Equal Variances: *MZ* versus Organization; Standard; Chapter.

| Test for Equal Variances | *MZ* vs. Organization; Standard; Chapter | | *MU* vs. Organization; Standard; Chapter | |
|---|---|---|---|---|
| Method | Test Statistic | *p*-Value | Test Statistic | *p*-Value |
| Multiple comparisons | — | 0.002 | — | 0.042 |
| Levene | 0.51 | 1.000 | 0.66 | 0.993 |

Comment: Null hypothesis—All variances are equal; Alternative hypothesis—At least one variance is different; Significance level $\alpha = 0.05$.

**Table 6.** Regression Equation.

| Variances | | |
|---|---|---|
| *MZ* | = | 24.354 − 1.03 Organization_AUT1 − 1.04 Organization_AUT2 + 6.55 Organization_AUT3<br>+ 13.27 Organization_OC − 17.75 Organization_TU + 3.05 Standard_EMS<br>− 9.35 Standard_EnMS + 12.81 Standard_IATF + 4.17 Standard_OHSAS − 10.68 Standard_QMS<br>− 1.73 Chapter_4 − 2.14 Chapter_5 − 3.60 Chapter_6 + 7.26 Chapter_7 + 4.15 Chapter_8<br>+ 0.56 Chapter_9 − 4.50 Chapter_10 |
| *MU* | = | 20,404 + 2.49 Organization_AUT1 + 1.31 Organization_AUT2 + 7.93 Organization_AUT3<br>+ 7.09 Organization_OC − 18.82 Organization_TU + 3.44 Standard_EMS<br>− 8.82 Standard_EnMS + 12.89 Standard_IATF + 5.12 Standard_OHSAS − 12.63 Standard_QMS<br>− 0.47 Chapter_4 − 3.08 Chapter_5 − 2.88 Chapter_6 + 6.42 Chapter_7 + 3.97 Chapter_8<br>− 0.05 Chapter_9 − 3.92 Chapter_10 |
| *Total* | = | 44.76 + 1.45 Organization_AUT1 + 0.28 Organization_AUT2 + 14.48 Organization_AUT3<br>+ 20.36 Organization_OC − 36.57 Organization_TU + 6.50 Standard_EMS<br>− 18.18 Standard_EnMS + 25.70 Standard_IATF + 9.29 Standard_OHSAS<br>− 23.31 Standard_QMS − 2.20 Chapter_4 − 5.22 Chapter_5 − 6.48 Chapter_6<br>+ 13.68 Chapter_7 + 8.12 Chapter_8 + 0.51 Chapter_9 − 8.42 Chapter_10 |

The multiple comparisons method showed that we could reject the hypothesis that the variances of the individual response factors of *MZ* and *MU* were statistically equal, since the *p*-Value was smaller than the alpha level. Thus, the condition for using the ANOVA method was not met. However, Levene's test showed that the hypothesis of equal variances for both *MZ* and *MU* responses could not be rejected, which supported the possibility of using the ANOVA method. Thus, this was an ambiguous result. Further conditions for the applicability of the ANOVA method resulted from the analysis of residuals.

Figure 3 shows a simple residual analysis performed for the *MZ*, *MU* and *Total* responses. The models corresponding to the presented residuals are described in Table 6. The residual analysis for each response is presented in four graphs. The first "Normal Probability Plot of Residuals" shows the normality of the distribution of residuals. The next plot, "Residuals versus Fits", plots the residuals according to their magnitude and shows that the residuals have a constant variance. The next graph is a "Histogram of residuals" where we can see if the data is skewed or if there are outliers in the data. Finally, the "Residuals versus Order" is presented in the order in which they were recorded.

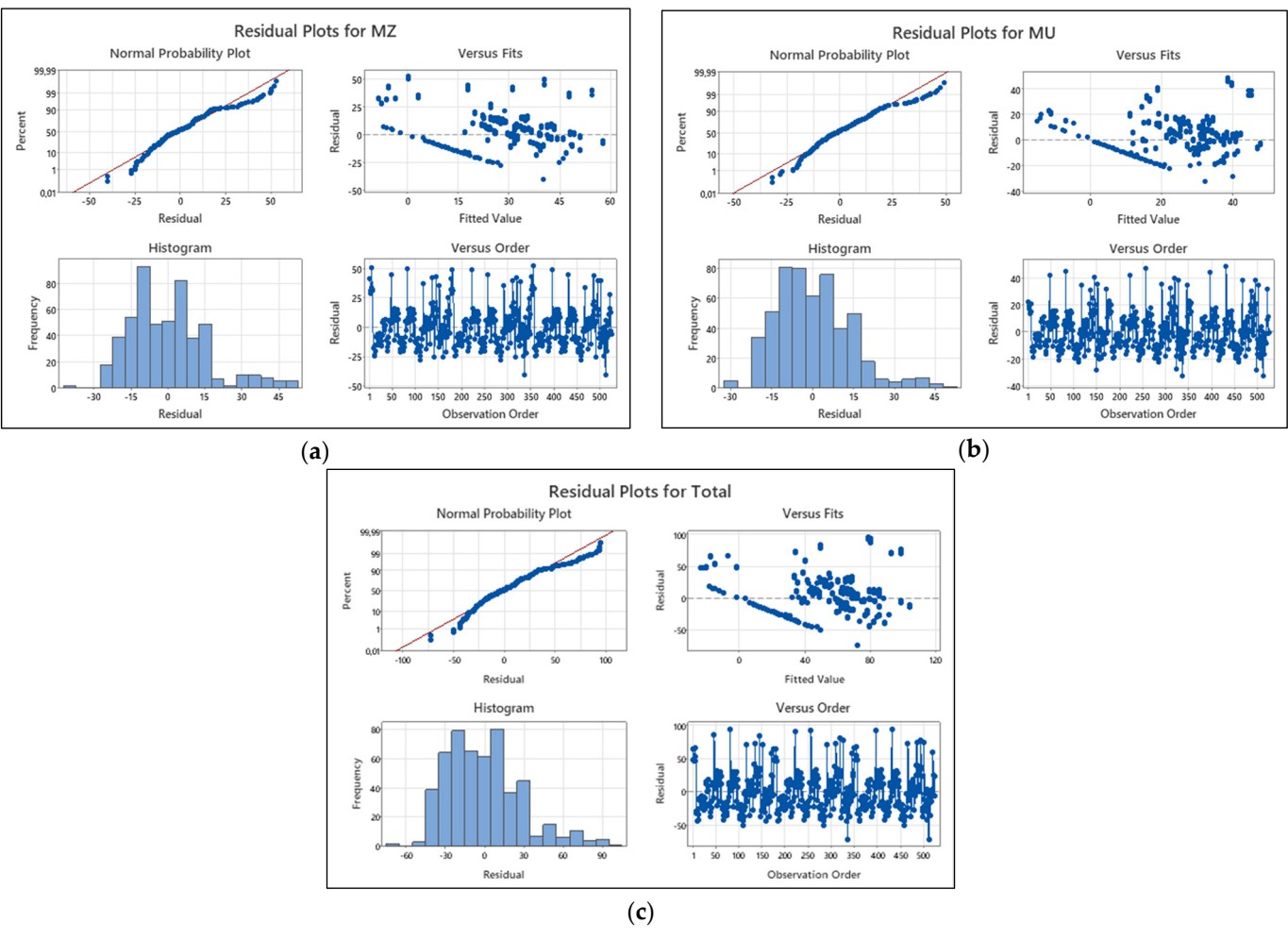

**Figure 3.** Residual plots for *MZ* (**a**), *MU* (**b**) and *Total* (**c**).

In the following, we constructed a pseudo linear regression (PLR) model as described in [89]. Due to its large size, we have presented only the main factors without interactions in Table 6.

The ANOVA then performed confirmed statistically significant differences for each level for all factors and all responses. This was confirmed by the low *p*-value shown as "0.000" in Table 7. On the other hand, the index of determination indicated that the model explained the *MZ* response at only 40.69%, the *MU* response at 48.32%, and *Total* at 45.32%.

Due to the controversial verification of homoskedasticity, we present the results of this method only informatively. In further validation, we focused on a procedure known as Bonferroni's method.

**Table 7.** Analysis of Variance for *MZ*, *MU* and *Total*.

| | | ANOVA for *MZ* | | | | | ANOVA for *MU* | | | | | ANOVA for *Total* | | | |
|---|---|---|---|---|---|---|---|---|---|---|---|---|---|---|---|
| **Source** | **DF** | **Adj SS** | **Adj MS** | **F-Value** | ***p*-Value** | **DF** | **Adj SS** | **Adj MS** | **F-Value** | ***p*-Value** | **DF** | **Adj SS** | **Adj MS** | **F-Value** | ***p*-Value** |
| Organization | 4 | 56,111 | 14,027.8 | 53.22 | 0.000 | 4 | 49,848 | 12,462.1 | 64.78 | 0.000 | 4 | 205,746 | 51,436.4 | 59.93 | 0.000 |
| Standard | 4 | 41,192 | 10,297.9 | 39.07 | 0.000 | 4 | 46,331 | 11,582.8 | 60.21 | 0.000 | 4 | 174,509 | 43,627.1 | 50.83 | 0.000 |
| Chapter | 6 | 8324 | 1387.3 | 5.26 | 0.000 | 6 | 6781 | 1130.1 | 5.87 | 0.000 | 6 | 29,880 | 4980.0 | 5.80 | 0.000 |
| Error | 509 | 134,150 | 263.6 | | | 509 | 97,912 | 192.4 | | | 509 | 436,889 | 858.3 | | |
| Lack-of-Fit | 160 | 133,902 | 836.9 | 1173.66 | 0.000 | 160 | 97,731 | 610.8 | 1174.43 | 0.000 | 160 | 436,477 | 2728.0 | 2309.63 | 0.000 |
| Pure Error | 349 | 249 | 0.7 | | | 349 | 182 | 0.5 | | | 349 | 412 | 1.2 | | |
| **Total** | **523** | **239,730** | | | | **523** | **200,813** | | | | **523** | **846,784** | | | |

| **Model** | **S** | **R-sq** | **R-sq (adj)** | **R-sq (pred)** | | **S** | **R-sq** | **R-sq (adj)** | **R-sq (pred)** | | **S** | **R-sq** | **R-sq (adj)** | **R-sq (pred)** |
|---|---|---|---|---|---|---|---|---|---|---|---|---|---|---|
| | 16.2344 | 44.04% | 42.50% | 40.69% | | 13.8695 | 51.24% | 49.90% | 48.32% | | 29.2972 | 48.41% | 46.99% | 45.32% |

In a post hoc ANOVA test [87,90–92] at 95% confidence intervals, it is evident that the results are completely independent of the results of the ANOVA used. This method does not require any specific assumptions about the dataset, as it is a multiple comparison correction that is used to control the overall level of Type I error. Table 8 clearly presents the groupings of the individual factor levels. The unequal position of some levels could already be observed with different model coefficients, but using a post hoc ANOVA test and Bonferroni analysis, these clusters were clearly determined.

**Table 8.** Grouping Information Using the Bonferroni Method and 95% Confidence.

| | | *MZ* | | | | | *MU* | | | | | *Total* | | | |
|---|---|---|---|---|---|---|---|---|---|---|---|---|---|---|---|
| **Organization** | **N** | **Mean** | **Grouping** | | | **Mean** | **Grouping** | | | | **Mean** | **Grouping** | | | |
| OC | 104 | 37.6238 | A | | | 27.4940 | A | B | | | 65.1178 | A | | | |
| AUT3 | 105 | 30.9007 | | B | | 28.3330 | A | | | | 59.2337 | A | | | |
| AUT1 | 105 | 23.3214 | | | C | 22.8904 | | B | C | | 46.2118 | | B | | |
| AUT2 | 105 | 23.3170 | | | C | 21.7191 | | | C | | 45.0362 | | B | | |
| TU | 105 | 6.6053 | | | | D | 1.5842 | | | | D | 8.1895 | | | C |

| **Standard** | **N** | **Mean** | **Grouping** | | | **Mean** | **Grouping** | | | **Mean** | **Grouping** | | |
|---|---|---|---|---|---|---|---|---|---|---|---|---|---|
| IATF | 105 | 37.1635 | A | | | 33.2922 | A | | | 70.4557 | A | | |
| OHSAS | 104 | 28.5268 | | B | | 25.5229 | | B | | 54.0497 | | B | |
| EMS | 105 | 27.4081 | | B | | 23.8490 | | B | | 51.2570 | | B | |
| EnMS | 105 | 14.9998 | | | C | 11.5825 | | | C | 26.5823 | | | C |
| QMS | 105 | 13.6701 | | | C | 7.7743 | | | C | 21.4444 | | | C |

| **Chapter** | **N** | **Mean** | **Grouping** | | | **Mean** | **Grouping** | | | **Mean** | **Grouping** | | |
|---|---|---|---|---|---|---|---|---|---|---|---|---|---|
| 7 | 75 | 31.6125 | A | | | 26.8287 | A | | | 58.4412 | A | | |
| 8 | 75 | 28.5033 | A | B | | 24.3751 | A | B | | 52.8784 | A | B | |
| 9 | 75 | 24.9143 | A | B | C | 20.3585 | A | B | C | 45.2728 | A | B | C |
| 4 | 74 | 22.6282 | | B | C | 19.9340 | A | B | C | 42.5622 | | B | C |
| 5 | 75 | 22.2141 | | B | C | 17.3245 | | | C | 39.5387 | | B | C |
| 6 | 75 | 20.7540 | | B | C | 17.5241 | | B | C | 38.2781 | | B | C |
| 10 | 75 | 19.8492 | | | C | 16.4841 | | | C | 36.3333 | | | C |

*N*—number of responses for individual factor levels; A, B, C, D—Bonferroni Significance Grouping.

For clarity, we presented the results in detail using Bonferroni Simultaneous 95% CIs (confidence intervals) for pairwise differences in means (see Figure 4). The present confidence intervals for pairwise differences of the means of individual levels were performed

separately for *MZ* response, *MU* response, and *Total*. We also conducted the analysis for each Organization, Standard, and Chapter factor separately.

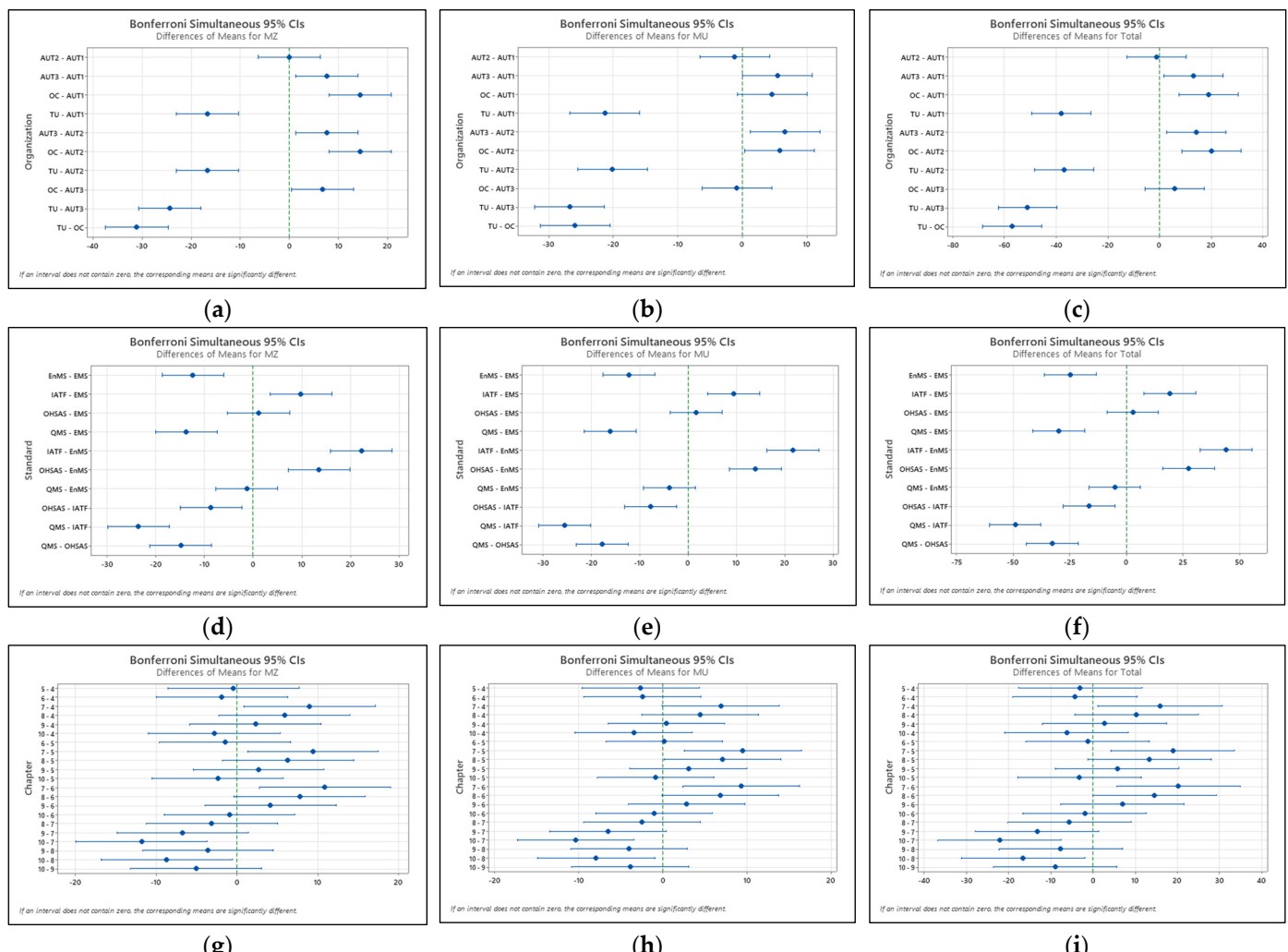

**Figure 4.** Plots by Bonferroni Simultaneous 95% CIs. Response for Organization *MZ* (**a**), *MU* (**b**) and *Total* (**c**); Response for Standard *MZ* (**d**), *MU* (**e**) and *Total* (**f**); Responsible for Chapter *MZ* (**g**), *MU* (**h**) and *Total* (**i**).

Those confidence intervals that contain a value of 0 represent non-significant differences between the means of the two levels. Intervals that do not contain a value of 0 indicate significant differences between the means of the two levels. In the case of the *MZ* response factor Organization (see Figure 4a), it is evident that there is one confidence interval that contains a value 0 and that is between AUT2–AUT1 organizations, which represents an insignificant difference. However, for the *MU* response (see Figure 4b) with the same factor, there are three confidence intervals that contain a value 0. This is a non-significant pairwise difference in means between the AUT3–AUT1, OC–AUT1 and OC–AUT3 organizations. In the Total graph (see Figure 4c), the value 0 occurs only for the confidence intervals of the AUT2–AUT1 and OC–AUT3 organizations.

In the graphs (see Figure 4), it is also possible to see confidence intervals that contain only negative values, which means that there is a statistically significant difference between the means, with the first mean being smaller than the second mean. In the case of the *MZ* response for the Standard factor (see Figure 4d), it is possible to see a statistically significant pairwise difference of means between EnMS–EMS, QMS–EMS, OHSAS–IATF, QMS–IATF, and QMS–OHSAS. The same is true for the *MU* and *Total* response.

Similarly, in Figure 4, confidence intervals containing only positive values, meaning that there is a statistically significant difference between the means and the first mean is larger than the second, are also evident according to the *MU* and *Total* response for the Chapter factor (see Figure 4g, i). There is a statistically significant pairwise difference in means between Chapters 7-4, 7-5 and 7-6, with Chapters 7-5, 8-5 and 7-6 for the *MU* response (see Figure 4h).

## 5. Discussion

Using Bonferroni's method, a relatively well-explained PLR methodology was developed. The coefficients for each factor (Organization, Standard, and Chapter) accurately determine the differences in the ratings of the two *MZ* and *MU* responses as a function of the factor levels. The main effects graphs for the *MZ* and *MU* responses and Total describe the situation very clearly (see Figure 5).

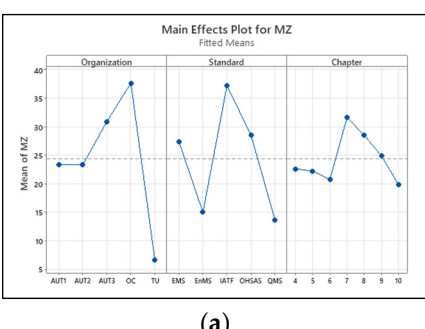

(a)

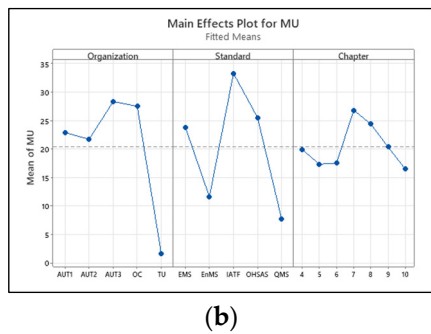

(b)

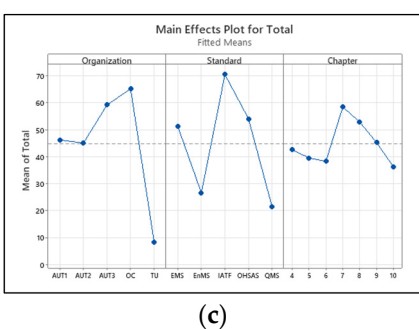

(c)

**Figure 5.** Main Effects Plot for *MZ* (**a**), *MU* (**b**) and *Total* (**c**).

In the *MZ* response and the evaluation of the Organization factor (see Figure 6a), an even position can be seen, with relatively little change for AUT1 and AUT2 organizations. The other organizations form separate groups (see also Table 8). For the *MU* response (see Figure 6b), it can be seen that AUT1 and AUT2 belong to the same group, while AUT2 and OC, OC and AUT 1 form different groups. In the case of the Standard evaluation, except for some shift, the position of all considered standards for the *MU*, *MZ*, and *Total* responses (see Figure 5) is very similar. The IATF standard has its own position. The OHS and EMS Standards are very similar, as well as the EnMS and QMS Standards. The above-mentioned similarity in the rankings of the individual standards is also evident from the Bonferroni simultaneous 95% CIs (see Figure 4d–f). When evaluating the individual chapters, except for Chapters 4 and 5, a high similarity of evaluation can also be noted.

As with the main factors, the individual interactions for the *MZ* and *MU* response ((and hence *Total*) were also graphed. For the *MZ* response (see Figure 6a), groups of chapters 7, 8, and 9 are formed, and for *MU* (see Figure 6b), Chapter 4 is added. The second group for *MZ* consists of all chapters except 7 and 10, and for *MU* it is Chapters 8, 9, 4, and 6. The last group for *MZ* and *MU* responses consists of all but Chapters 7 and 8.

The above-mentioned similarity is also evident from the Bonferroni simultaneous 95% CIs (see Figure 4g–i), which in most cases contain a null value.

While the low ranking of TU was expected given the number of standards (MSs) implemented, it is also interesting to note the large difference between the EnMS or QMS standards and the other three, with the IATF standard achieving the highest ranking and therefore the largest difference in ranking. Similarly, but not so differently, in the chapter-by-chapter ratings, Chapters 7 and 9 have significantly higher ratings than the other chapters for both the *MZ* and *MU* responses. When evaluating the interactions in both cases, the "chapter-enterprise" interaction is relatively insignificant (see Figure 6c).

A more significant interaction can be observed in the case of "chapter and standard", which is probably due to the significant position of Chapter 8 for the IATF standard. Relatively more significant interactions were recorded from the "standard and organization"

perspective. However, the given situation can be interpreted as the absence of an evaluation of some standards by those organizations that have not implemented them.

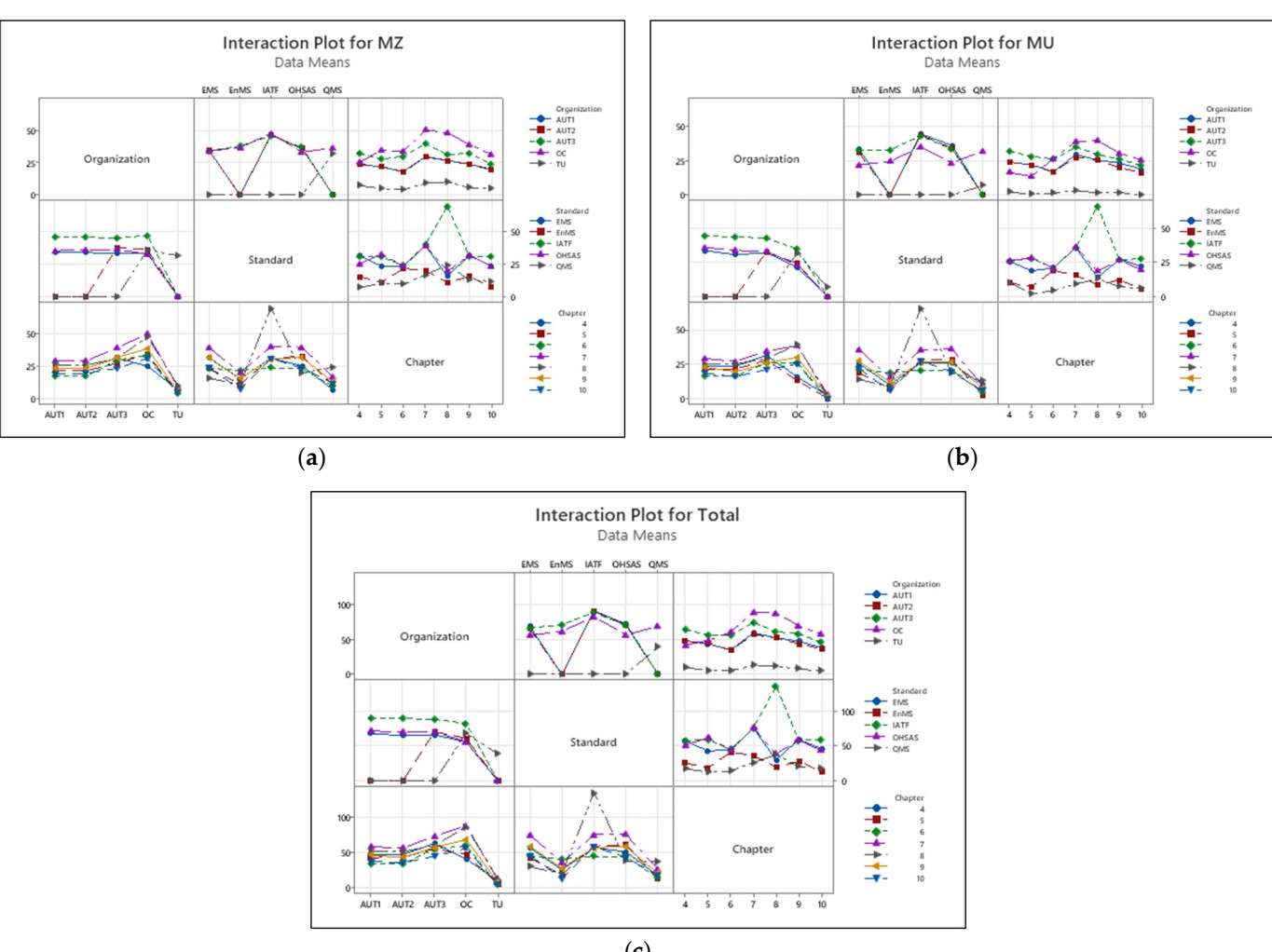

**Figure 6.** Interaction Plot for *MZ* (**a**), *MU* (**b**) and *Total* (**c**).

## 6. Conclusions

The application of the Bonferroni method confirmed the hypotheses that the developed SRIMS model is a sufficiently appropriate tool for assessing the overall level of applicability of CSR requirements to established MSs. For the future development of this tool, its extension to other standardized requirements and codes is being considered; this may also lead to a change in its calibration based on the structure of CSR requirements for different suppliers. The development of a more appropriate application tool (software) will allow us to extend the use of this tool to other MSs and thus also to extend the scope of its use to other (e.g., food) industries [44]. The advantage of the initial SRIMS model is that it allows us to assess the efficiency of MSs management itself also with regard to CSR requirements. When the model is complemented by the methodology of assessing the level of integration based on risk-based thinking [93,94], it can be transformed into an effective tool for assessing even the level of integration of MSs themselves and evaluating their level of effectiveness for specific CSR requirements. The aim of such a model is not only to support the fulfilment of the certification requirements of MSs but to create a self-assessment tool that enables the management of the supplier organization to react flexibly to changes in customer requirements. Another criterion would be to establish a framework for assessing the level achieved (e.g., poor level, medium to excellent level),

thus this model would also set an incentive framework for improvement in those areas that are key for achieving the business objectives of the management of the organization.

The SRIMS methodology has demonstrated the necessity of measuring and assessing the level of applicability of CSR in MSs in organizations operating in different industries. Its application has verified the acceptability of this methodology as an effective tool for top management´s decision making. SRIMS has created a prerequisite for the emergence of a supportive comprehensive tool for assessing various CSR requirements and selecting appropriate strategic tools for business development in the current global environment.

The objective of this paper was to analyze the different CSR approaches in internationally recognized cross-industry and industry specific standards and codes in different industries. A major challenge for the future is to analyze the above CSR approaches in those industrial enterprises that significantly implement the principles of the method and paradigm known as Industry 4.0.

**Author Contributions:** Conceptualization, H.P. and R.T.; methodology, F.K. and H.P.; validation, F.K., A.S. and P.D.; formal analysis, F.K. and R.T; resources, F.K. and P.D.; writing—review and editing, A.S.; supervision, H.P. All authors have read and agreed to the published version of the manuscript.

**Funding:** This contribution is the result of the projects implementation: APVV No. APVV-19-0367 Framework of the Integrated Process Safety Management Approach for the Intelligent Enterprise, and Research and development of intelligent traumatological external fixation systems manufactured by digitalization methods and additive manufacturing technology (Acronym: SMARTfix), ITMS2014+: 313011BWQ1 supported by the Operational Program Integrated Infrastructure funded by the European Regional Development Fund.

**Institutional Review Board Statement:** Not applicable.

**Informed Consent Statement:** Not applicable.

**Data Availability Statement:** Not applicable.

**Conflicts of Interest:** The authors declare no conflict of interest.

## Appendix A

**Table A1.** Determination of the weights ($W_{X.Y}$) within the methodological framework.

| Standard/Chapters | | Management System (MS) | | | | | | | | | | | |
|---|---|---|---|---|---|---|---|---|---|---|---|---|---|
| | | IQNet SR10 | | QMS | | IATF | | EMS | | OHSMS | | EnMS | |
| | | Chapt. | $W_{X.Y}$ | Chapt. | $W_{X.Y}$ | Chapt. | $W_{X.Y}$ | Chapt. | $W_{X.Y}$ | Chapt. | $W_{X.Y}$ | Chapt. | $W_{X.Y}$ |
| 4 | Context of the organization | 4.1 | 20 | 4.1 | 20 | 4.1 | 20 | 4.1 | 20 | 4.1 | 20 | 4.1 | 20 |
| | | 4.2 | 20 | 4.2 | 20 | 4.2 | 20 | 4.2 | 20 | 4.2 | 20 | 4.2 | 20 |
| | | 4.3 | 20 | 4.3 | 20 | 4.3 | 20 | 4.3 | 20 | 4.3 | 20 | 4.3 | 20 |
| | | 4.4 | 20 | 4.4 | 20 | 4.4 | 20 | 4.4 | 20 | 4.4 | 20 | 4.4 | 20 |
| 5 | Leadership | 5.1 | 20 | 5.1 | 20 | 5.1 | 20 | 5.1 | 20 | 5.1 | 20 | 5.1 | 20 |
| | | 5.2 | 20 | 5.2 | 20 | 5.2 | 20 | 5.2 | 20 | 5.2 | 20 | 5.2 | 20 |
| | | 5.3 | 20 | 5.3 | 20 | 5.3 | 20 | 5.3 | 20 | 5.3 | 20 | 5.3 | 20 |
| | | 5.4 | 20 | | | 5.1.1.1 | 20 | | | 5.4 | 20 | | |
| 6 | Planning | 6.1 | 20 | 6.1 | 20 | 6.1 | 20 | 6.1 | 20 | 6.1 | 20 | 6.1 | 20 |
| | | 6.2 | 20 | 6.2 | 20 | 6.2 | 20 | 6.2 | 20 | 6.2 | 20 | 6.2 | 20 |
| | | 6.3 | 20 | 6.3 | 20 | 6.3 | 20 | | | | | 6.3 | 20 |
| | | 6.4 | 20 | | | | | 6.1.3 | 20 | 6.1.3 | 20 | 6.4 | 20 |
| | | | | | | | | | | | | 6.5 | 20 |
| | | | | | | | | | | | | 6.6 | 20 |

**Table A1.** *Cont.*

| Standard/Chapters | | IQNet SR10 | | QMS | | IATF | | EMS | | OHSMS | | EnMS | |
|---|---|---|---|---|---|---|---|---|---|---|---|---|---|
| | | Chapt. | $W_{X.Y}$ | Chapt. | $W_{X.Y}$ | Chapt. | $W_{X.Y}$ | Chapt. | $W_{X.Y}$ | Chapt. | $W_{X.Y}$ | Chapt. | $W_{X.Y}$ |
| 7 | Support | 7.1 | 20 | 5.1 | 20 | 5.1 | 20 | 5.1 | 20 | 5.1 | 20 | 5.1 | 20 |
| | | 7.2 | 20 | 5.2 | 20 | 5.2 | 20 | 5.2 | 20 | 5.2 | 20 | 5.2 | 20 |
| | | 7.3 | 20 | 5.3 | 20 | 5.3 | 20 | 5.3 | 20 | 5.3 | 20 | 5.3 | 20 |
| | | 7.4 | 20 | 5.4 | 20 | 5.4 | 20 | 5.4 | 20 | 5.4 | 20 | 5.4 | 20 |
| | | 7.5 | 20 | 5.3 | 20 | 5.3 | 20 | 5.3 | 20 | 5.3 | 20 | 5.3 | 20 |
| 8 | Operation | 8.1 | 20 | 8.1 | 20 | 8.1 | 30 | 8.1 | 20 | 8.1 | 30 | 8.1 | 20 |
| | | 8.2 | 20 | 8.2 | 20 | 8.2 | 20 | 8.2 | 20 | 8.2 | 20 | 8.2 | 20 |
| | | 8.3 | 30 | 8.3 | 30 | 8.3 | 40 | | | | | 8.3 | 20 |
| | | 8.4 | 20 | 8.4 | 20 | 8.4 | 20 | | | | | | |
| | | 8.5 | 20 | 8.5 | 20 | 8.5 | 20 | | | | | | |
| | | 8.6 | 20 | 8.6 | 20 | 8.6 | 30 | | | | | | |
| | | 8.7 | 20 | 8.7 | 20 | 8.7 | 20 | | | | | | |
| | | 8.8 | 20 | | | | | | | | | | |
| | | 8.9 | 20 | | | | | | | | | | |
| 9 | | 9.1 | 40 | 9.1 | 40 | 9.1 | 40 | 9.1 | 40 | 9.1 | 40 | 9.1 | 40 |
| | Performance evaluation | 9.2 | 20 | 9.2 | 20 | 9.2 | 20 | 9.2 | 20 | 9.2 | 20 | 9.2 | 20 |
| | | 9.3 | 20 | 9.3 | 20 | 9.3 | 20 | 9.3 | 20 | 9.3 | 20 | 9.3 | 20 |
| | | 9.4 | 20 | | | | | | | | | | |
| 10 | Improvement | 10.1 | 40 | 10.1 | 40 | 10.1 | 40 | 10.1 | 40 | 10.1 | 40 | 10.1 | 40 |
| | | 10.2 | 40 | 10.2 | 40 | 10.2 | 40 | 10.2 | 40 | 10.2 | 40 | 10.2 | 40 |
| | In total $W_X$ | | 710 | | 610 | | 660 | | 500 | | 540 | | 580 |

## Appendix B

**Table A2.** SRIMS application in TU.

| | Organization: TU | | | | |
|---|---|---|---|---|---|
| Standard/Chapters | Management System (MS) | | | | |
| | IQNet SR10 | QMS | EMS | OHSMS | EnMS |
| CHAP4 | 80 | 47 | 0 | 0 | 0 |
| CHAP5 | 80 | 28 | 0 | 0 | 0 |
| CHAP6 | 80 | 27 | 0 | 0 | 0 |
| CHAP7 | 100 | 46 | 0 | 0 | 0 |
| CHAP8 | 190 | 59 | 0 | 0 | 0 |
| CHAP9 | 100 | 38 | 0 | 0 | 0 |
| CHAP10 | 80 | 24 | 0 | 0 | 0 |
| MS/IQNet SR10 % | | 38% | 0% | 0% | 0% |
| In total $W_X$ | 710 | 269 | 0 | 0 | 0 |

**Table A3.** SRIMS application in AU1.

| Standard/Chapters | Organization: AU1 | | | | |
| | Management System (MS) | | | | |
| | IQNet SR10 | IATF | EMS | OHSMS | EnMS |
|---|---|---|---|---|---|
| CHAP4 | 80 | 80 | 80 | 80 | 0 |
| CHAP5 | 80 | 80 | 60 | 80 | 0 |
| CHAP6 | 80 | 57 | 59 | 59 | 0 |
| CHAP7 | 100 | 96 | 99 | 100 | 0 |
| CHAP8 | 190 | 167 | 40 | 50 | 0 |
| CHAP9 | 100 | 78 | 79 | 79 | 0 |
| CHAP10 | 80 | 70 | 59 | 59 | 0 |
| MS/IQNet SR10 % | | 88% | 67% | 71% | 0% |
| In total $W_X$ | 710 | 628 | 476 | 507 | 0 |

**Table A4.** SRIMS application in AU2.

| Standard/Chapters | Organization: AU2 | | | | |
| | Management System (MS) | | | | |
| | IQNet SR10 | IATF | EMS | OHSMS | EnMS |
|---|---|---|---|---|---|
| CHAP4 | 80 | 80 | 80 | 80 | 0 |
| CHAP5 | 80 | 80 | 60 | 80 | 0 |
| CHAP6 | 80 | 57 | 58 | 59 | 0 |
| CHAP7 | 100 | 93 | 94 | 98 | 0 |
| CHAP8 | 190 | 165 | 38 | 49 | 0 |
| CHAP9 | 100 | 74 | 73 | 73 | 0 |
| CHAP10 | 80 | 65 | 54 | 54 | 0 |
| MS/IQNet SR10 % | | 86% | 64% | 69% | 0% |
| In total $W_X$ | 710 | 614 | 457 | 493 | 0 |

**Table A5.** SRIMS application in AU3.

| Standard/Chapters | Organization: AU3 | | | | |
| | Management System (MS) | | | | |
| | IQNet SR10 | IATF | EMS | OHSMS | EnMS |
|---|---|---|---|---|---|
| CHAP4 | 80 | 80 | 80 | 80 | 80 |
| CHAP5 | 80 | 80 | 60 | 80 | 60 |
| CHAP6 | 80 | 57 | 58 | 58 | 108 |
| CHAP7 | 100 | 93 | 94 | 94 | 92 |
| CHAP8 | 190 | 165 | 38 | 49 | 52 |
| CHAP9 | 100 | 72 | 74 | 72 | 71 |
| CHAP10 | 80 | 65 | 54 | 54 | 35 |
| MS/IQNet SR10 % | | 86% | 65% | 69% | 70% |
| In total $W_X$ | 710 | 612 | 458 | 487 | 498 |

**Table A6.** SRIMS application in OC.

| Standard/Chapters | Organization: OC | | | | |
|---|---|---|---|---|---|
| | Management SYSTEM (MS) | | | | |
| | IQNet SR10 | IATF | EMS | OHSMS | EnMS |
| CHAP4 | 80 | 52 | 52 | 52 | 52 |
| CHAP5 | 80 | 59 | 36 | 71 | 36 |
| CHAP6 | 80 | 55 | 51 | 51 | 108 |
| CHAP7 | 100 | 91 | 90 | 88 | 86 |
| CHAP8 | 190 | 161 | 36 | 45 | 50 |
| CHAP9 | 100 | 66 | 72 | 72 | 72 |
| CHAP10 | 80 | 63 | 52 | 52 | 34 |
| MS/IQNet SR10 % | | 77% | 55% | 61% | 62% |
| In total $W_X$ | 710 | 547 | 389 | 431 | 438 |

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
