# Peer review of "The Methodology for Assessing the Applicability of CSR into Supplier Management Systems"

_sustainability, doi:10.3390/su151713240_

Round 1

Reviewer 1 Report

The paper is interesting. However, it has some problems as follows:

1、  Please rewrite the abstract to present the main idea of the paper.

2、  The drawback of the current research is not fully discussed so that the motivation is not very clear.

3、  What are the main contributions of the paper? Please list them one by one.

4、  What’s the main limitation of the work? Please also highlight the future work.

5、  What’s the main relationship between Management Systems and CSR?

6、  Why do you discuss the following three sectors: automotive industry, research organization, and metallurgical industry? What are the main differences?

The Quality of English Language should be improved. 

Author Response

Thank you for all the recommendations and comments. All grammatical changes are marked by green color in the text. The manuscript has been revised by a native speaker.

We marked additions or modifications to the manuscript in yellow (see the attached document).

Reviewer 2 Report

(1). Abstract: Reads well.

(2). Keywords: Add one more appropriate keyword

(3). Introduction: This section messes that why this particular study is needed, what gaps will be addressed by doing this research and what will be the contribution of the research, research questions. I suggest a complete re-writing of Introduction section considering the points raised.

(4). Theoretical Framework: This is another weak section of the study. To demonstrate an adequate understanding of the relevant literature in the field and cite an appropriate range of literature sources. I suggest a complete re-writing of literature review section considering the systematic literature review.

(5). Materials and Methods: Elaborate the details of the research methodology. Justify, why adopted this approach. Discus the suitability of this methodology for this problem.  Address the advantages of this methodology over other methodologies. Add demographic profile of respondents.

(6). Results: Add more interpretations and highlight the novel aspects

(7). Implications and Contributions: Further discuss the theoretical and practical implications of this study

(8). Conclusion: It should be enhancing the contributions, limitations, underscore the scientific value added, and/or the applicability of your findings/results.

(9). Citations/references are not proper format in the entire paper. It should be corrected. Attention should be paid to clarity of expression and readability

Attention should be paid to clarity of expression and readability

Author Response

(The authors gave the same response as above.)

Reviewer 3 Report

The topic of the paper is interesting and up-to-date. But the paper have many flaws nedds to resolve:

The text seems to lack clear transitions between different sections and points, which can make it difficult for readers to follow the flow of ideas. A well-organized structure and effective use of headings/subheadings can help improve readability.

While the excerpt provides graphs and figures, the interpretation of these results is limited. There's a need for more detailed explanations of the patterns observed in the graphs and how they align with the research questions or hypotheses.

The excerpt mentions "the necessity of measuring and assessing the level of applicability of CSR in MSs in organizations operating in different industries." However, the actual comparison between different industries and their CSR approaches is not detailed enough.

It's not entirely clear what theoretical framework or literature review underpins the research. A well-defined theoretical foundation helps readers understand the context and motivation of the study.

The paper could benefit from a more explicit discussion of how the results can be practically applied or what recommendations can be drawn from the study.

The introduction could be expanded to provide more context on the SRIMS methodology and its relevance in the field of sustainability assessment. It should clearly state the research questions and objectives.

Elaborate on the criteria used to select organizations for the study, including the rationale behind their selection. This will enhance the transparency of your research approach.

Ensure that all graphs are properly labeled with clear titles and axis labels. Include legends that explain the meaning of different colors or symbols in the graphs.

Go beyond surface-level observations. Analyze and interpret the patterns and trends shown in the graphs. Discuss the practical implications of these findings for each organization, standard, and chapter.

Clearly articulate the managerial implications of your findings. Provide actionable recommendations for businesses to enhance their CSR strategies based on the study's outcomes.

The text appears to have some grammatical issues and unclear sentences. A well-written paper should be free from linguistic errors to ensure effective communication.

Author Response

(The authors gave the same response as above.)

Round 2

Reviewer 1 Report

The paper is revised well and the introduction can be further improved. 

The Quality of English Language can be enhanced. 

Author Response

First of all, thank you for your comments on the article.

We are forwarding the requested corrections. In the introduction, we added - lines 55-67.

We marked minor grammatical errors and corrections directly in the text in purple.

Reviewer 3 Report

The Authors made the good job to improve the etxt according to remarks.

It's ok.

Author Response

(The authors gave the same response as above.)
